# Predicting Training Time Without Training

**Luca Zancato**
Department of Information Engineering
University of Padova
luca.zancato@phd.unipd.it

**Alessandro Achille**
Amazon Web Services
aachille@amazon.com

**Avinash Ravichandran**
Amazon Web Services
ravinash@amazon.com

**Rahul Bhotika**
Amazon Web Services
bhotikar@amazon.com

**Stefano Soatto**
Amazon Web Services
soattos@amazon.com

## Abstract

We tackle the problem of predicting the number of optimization steps that a pre-trained deep network needs to converge to a given value of the loss function. To do so, we leverage the fact that the training dynamics of a deep network during fine-tuning are well approximated by those of a linearized model. This allows us to approximate the training loss and accuracy at any point during training by solving a low-dimensional Stochastic Differential Equation (SDE) in function space. Using this result, we are able to predict the time it takes for Stochastic Gradient Descent (SGD) to fine-tune a model to a given loss without having to perform any training. In our experiments, we are able to predict training time of a ResNet within a 20% error margin on a variety of datasets and hyper-parameters, at a 30 to 45-fold reduction in cost compared to actual training. We also discuss how to further reduce the computational and memory cost of our method, and in particular we show that by exploiting the spectral properties of the gradients' matrix it is possible to predict training time on a large dataset while processing only a subset of the samples.

## 1 Introduction

Say you are a researcher with many more ideas than available time and compute resources to test them. You are pondering to launch thousands of experiments but, as the deadline approaches, you wonder whether they will finish in time, and before your computational budget is exhausted. Could you predict the time it takes for a network to converge, before even starting to train it?

We look to efficiently estimate the number of training steps a Deep Neural Network (DNN) needs to converge to a given value of the loss function, without actually having to train the network. This problem has received little attention thus far, possibly due to the fact that the initial training dynamics of a randomly initialized DNN are highly non-trivial to characterize and analyze. However, in most practical applications, it is common to *not* start from scratch, but from a pre-trained model. This may simplify the analysis, since the final solution obtained by fine-tuning is typically not too far from the initial solution obtained after pre-training. In fact, it is known that the dynamics of overparametrized DNNs [9, 32, 2] during fine-tuning tends to be more predictable and close to convex [25]. Our main contribution is to introduce the problem of predicting training time in realistic use cases, in particular how training time depends on the hyper-parameters and, most importantly, on the interaction between target task and pre-training (which, to the best of our knowledge, is new).

We therefore characterize the training dynamics of a pre-trained network and provide a computationally efficient procedure to estimate the expected profile of the loss curve over time.

We use a linearized version of the DNN model around pre-trained weights to study its actual dynamics. In [21] a similar technique is used to describe the learning trajectories of *randomly initialized* wide neural networks. Such an approach is inspired by the Neural Tangent Kernel (NTK) for infinitely wide networks [14]. While we note that NTK theory may not correctly predict the dynamics of real (finite size) randomly initialized networks [12], we show that our linearized approach can be extended to fine-tuning of real networks in a similar vein to [25]. In order to predict fine-tuning Training Time (TT) without training we introduce a Stochastic Differential Equation (SDE) (similar to [13]) to approximate the behavior of SGD: we do so for a linearized DNN and in function space rather than in weight space. That is, rather than trying to predict the evolution of the weights of the network (a $D$-dimensional vector), we aim to predict the evolution of the outputs of the network on the training set (a $N \times C$-dimensional vector, where $N$ is the size of the dataset and $C$ the number of network's outputs). This drastically reduces the dimensionality of the problem for over-parametrized networks (that is, when $NC \ll D$).

A possible limiting factor of our approach is that the memory requirement to predict the dynamics scales as $O(DC^2N^2)$. This would rapidly become infeasible for datasets of moderate size and for real architectures ($D$ is in the order of millions). To mitigate this, we show that we can use random projections to restrict to a much smaller $D_0$-dimensional subspace with only minimal loss in prediction accuracy. We also show how to estimate Training Time using a small subset of $N_0$ samples, which reduces the total complexity to $O(D_0 \, C^2 N_0^2)$. We do this by exploiting the spectral properties of the Gram matrix of the gradients. Under mild assumptions the same tools can be used to estimate Training Time on a larger dataset without actually seeing the data.

To summarize, our main contributions are:

(i) We present both a qualitative and quantitative analysis of the fine-tuning Training Time as a function of the Gram-Matrix $\Theta$ of the gradients at initialization (empirical NTK matrix).

(ii) We show how to reduce the cost of estimating the matrix $\Theta$ using random projections of the gradients, which makes the method efficient for common architectures and large datasets.

(iii) We introduce a method to estimate how much longer a network will need to train if we increase the size of the dataset without actually having to see the data (under the hypothesis that new data is sampled from the same distribution).

(iv) We test the accuracy of our predictions on off-the-shelf state-of-the-art models trained on real datasets. We are able to predict the correct training time within a 20% error with 95% confidence over several different datasets and hyperparameters at only a small fraction of the time it would require to actually run the training (30-45x faster in our experiments).

## 2   Related Work

Predicting the training time of a state-of-the-art architecture on large scale datasets is a relatively understudied topic. In this direction, Justus et al. [15] try to estimate the wall-clock time required for a forward and backward pass on given hardware. We focus instead on a complementary aspect: estimating the number of fine-tuning steps necessary for the loss to converge below a given threshold. Once this has been estimated we can combine it with the average time for the forward and backward pass to get a final estimate of the wall clock time to fine-tune a DNN model without training it.

Hence, we are interested in predicting the learning dynamics of a pre-trained DNN trained with either Gradient Descent (GD) or Stochastic Gradient Descent (SGD).

In order to predict the learning dynamics many works are based on learning curve prediction (see [17] and references therein). These methods mainly focus on predicting the effect of different hyper-parameters for fixed task and architectures. Differently, we provide qualitative interpretation and quantitative prediction of the convergence speed of a DNN as a function of optimization hyper-parameters, network pre-training and target task.

Other results are known to describe training dynamics under a variety of assumptions (e.g. [16, 29, 27, 6]), in the following we are mainly interested on recent developments which describe the optimization dynamics of a DNN using a linearization approach. Several works [14, 20, 10] suggest that in the over-parametrized regime wide DNNs behave similar to linear models, and in particular

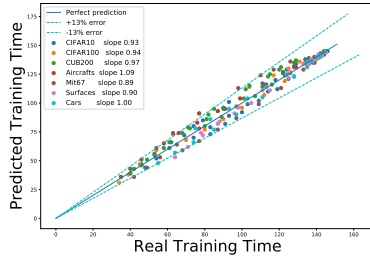
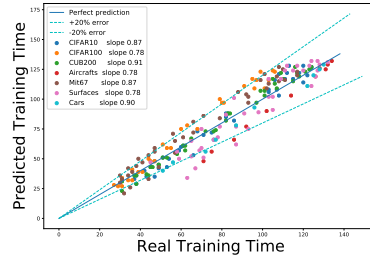

(a) Training with Gradient Descent.          (b) Training with SGD.

Figure 1: **Training time prediction (# iterations) for several fine-tuning tasks.** Scatter plots of the predicted time vs the actual training time when fine-tuning a ResNet-18 pre-trained on ImageNet on several tasks. Each task is obtained by randomly sampling a subset of five classes with 150 images (when possible) each from one popular dataset with different hyperparameters (batch size, learning rate). The closer the scatter plots to the bisector the better the TT estimate. Our prediction is **(a)** within 13% of the real training time 95% of the times when using GD and **(b)** within 20% of the real training time when using SGD.

they are fully characterized by the Gram-Matrix of the gradients, also known as empirical Neural Tangent Kernel (NTK).

Under these assumptions, [14, 3] derive a simple connection between training time and spectral decomposition of the NTK matrix. However, their results are limited to Gradient Descend dynamics and to simple architectures which are not directly applicable to real scenarios. In particular, their arguments hinge on the assumption of using a randomly initialized very wide two-layer or infinitely wide neural network [3, 11, 23]. We take this direction a step further, providing a unified framework which allows us to describe training time for both SGD and GD on common architectures.

Again, we rely on a linear approximation of the model, but while the practical validity of such linear approximation for randomly initialized state-of-the-art architectures (such as ResNets) is still discussed [12], we follow Mu et al. [25] and argue that the fine-tuning dynamics of over-parametrized DNNs can be closely described by a linearization. We expect such an approximation to hold true since the network does not move much in parameters space during fine-tuning and *over-parametrization* leads to smooth and regular loss function around the pre-trained weights [9, 32, 2, 22]. Under this premise, to tackle both GD and SGD in an unified framework we build on [13] modelling linearized training using a Stochastic Differential Equation in function space. We show we can use linearization to study the fine-tuning dynamics as suggested by [25] and provide accurate estimates on Training Time.

## 3   Predicting training time

In this section we look at how to efficiently approximate the training time of a DNN without actual training, to do so we use a linearized approximation of the DNN. Such an approximation can be computed without any fine-tuning and mimic the actual learning dynamics of the non-linear DNN. Therefore from the loss trajectory of the linearized model we can get an estimate of the actual Training Time of the non-linear model (see Supplementary Material for the complete algorithm).

By *Training Time* (TT) we mean the number of optimization steps – of either Gradient Descent (GD) or Stochastic Gradient Descent (SGD) – needed to bring the loss on the training set below a certain threshold.

We start by introducing our main tool. Let $f_w(x)$ denote the output of the network, where $w$ denotes the weights of the network and $x \in \mathbb{R}^d$ denotes its input (e.g., an image). Let $w_0$ be the weight configuration after pre-training. We assume that when fine-tuning a pre-trained network the solution remains close to pre-trained weights $w_0$ [25, 9, 32, 2]. Under this assumption – which we discuss further in Section 6 – we can faithfully approximate the network with its Taylor expansion around $w_0$ [21]. Let $w_t$ be the fine-tuned weights at time $t$. Using big-O notation and $f_t \equiv f_{w_t}$, we have:

$$f_t(x) = f_0(x) + \nabla_w f_0(x)|_{w=w_0}(w_t - w_0) + O(\|w_t - w_0\|^2)$$

We now want to use this approximation to characterize the training dynamics of the network during fine-tuning to estimate TT. In such theoretical analyses [14, 21, 3] it is common to assume that the network is trained with Gradient Descent (GD) rather than Stochastic Gradient Descent, and in the limit of a small learning rate. In this limit, the dynamics are approximated by the gradient flow differential equation $\dot{w}_t = -\eta \nabla_{w_t} \mathcal{L}$ [14, 21] where $\eta$ denotes the learning rate and $\mathcal{L}(w)$ denotes the loss function $\mathcal{L}(w) = \sum_{i=1}^{N} \ell(y_i, f_w(x_i))$., where $\ell$ is the per-sample loss function (e.g. Cross-Entropy). This approach however has two main drawbacks. First, it does not properly approximate Stochastic Gradient Descent, as it ignores the effect of the gradient noise on the dynamics, which affects both training time and generalization. Second, the differential equation involves the weights of the model, which live in a very high dimensional space thus making finding numerical solutions to the equation not tractable.

To address both problems, building on top of [13] in the Supplementary we prove the following result.

**Proposition 1** *In the limit of small learning rate $\eta$, the output on the training set of a linearized network $f_t^{lin}$ trained with SGD evolves according to the following Stochastic Differential Equation (SDE):*

$$df_t^{lin}(\mathcal{X}) = \underbrace{-\eta\Theta\nabla_{f_t^{lin}(\mathcal{X})}\mathcal{L}_t \, dt}_{\text{deterministic part}} + \underbrace{\frac{\eta}{\sqrt{|B|}}\nabla_w f_0^{lin}(\mathcal{X})\Sigma^{\frac{1}{2}}(f_t^{lin}(\mathcal{X}))dn}_{\text{stochastic part}}, \tag{1}$$

*where $\mathcal{X}$ is the set of training images, $|B|$ the batch-size and $dn$ is a $D$-dimensional Brownian motion. We have defined the Gram gradients matrix $\Theta$ [14, 28] (i.e., the empirical Neural Tangent Kernel matrix) and the covariance matrix $\Sigma$ of the gradients as follows:*

$$\Theta := \nabla_w f_0(\mathcal{X})\nabla_w f_0(\mathcal{X})^T, \tag{2}$$

$$\Sigma(f_t^{lin}(\mathcal{X})) := \mathbb{E}\big[(g_i\nabla_{f_t^{lin}(x_i)}\mathcal{L}) \otimes (g_i\nabla_{f_t^{lin}(x_i)}\mathcal{L})\big] - \mathbb{E}\big[g_i\nabla_{f_t^{lin}(x_i)}\mathcal{L}\big] \otimes \mathbb{E}\big[g_i\nabla_{f_t^{lin}(x_i)}\mathcal{L}\big]. \tag{3}$$

*where $g_i \equiv \nabla_w f_0(x_i)$. Note both $\Theta$ and $\Sigma$ only require gradients w.r.t. parameters computed at initialization.*

The first term of eq. (1) is an ordinary differential equation (ODE) describing the deterministic part of the optimization, while the second stochastic term accounts for the noise. In Figure 2 (left) we show the qualitative different behaviour of the solution to the deterministic part of eq. (1) and the complete SDE eq. (1). While several related results are known in the literature for the dynamics of the network in weight space [7], note that eq. (1) completely characterizes the training dynamics of the linearized model by looking at the evolution of the output $f_t^{lin}(\mathcal{X})$ of the model on the training samples – a $N \times C$-dimensional vector – rather than looking at the evolution of the weights $w_t$ – a $D$-dimensional vector. When the number of data points is much smaller than the number of weights (which are in the order of millions for ResNets), this can result in a drastic dimensionality reduction, which allows easy estimation of the solution to eq. (1). Solving eq. (1) still comes with some challenges, particularly in computing $\Theta$ efficiently on large datasets and architectures. We tackle these in Section 4. Before that, we take a look at how different hyper-parameters and different pre-trainings affect the training time of a DNN on a given task.

### 3.1 Effect of hyper-parameters on training time

**Effective learning rate.** From Proposition 1 we can gauge how hyper-parameters will affect the optimization process of the linearized model and, by proxy, of the original model it approximates. One thing that should be noted is that Proposition 1 assumes the network is trained with momentum $m = 0$. Using a non-zero momentum leads to a second order differential equation in weight space, that is not captured by Proposition 1. We can however, introduce heuristics to handle the effect of momentum: Smith et al. [29] note that the momentum acts on the stochastic part shrinking it by a factor $\sqrt{1/(1-m)}$. Meanwhile, under the assumptions we used in Proposition 1 (small learning rate), we can show (see Supplementary Material) the main effect of momentum on the deterministic part is to re-scale the learning rates by a factor $1/(1-m)$. Given these results, we define the effective learning rate (ELR) $\hat{\eta} = \eta/(1-m)$ and claim that, in first approximation, we can simulate the effect of momentum by using $\hat{\eta}$ instead of $\eta$ in eq. (1). In particular, models with different learning rates and momentum coefficients will have similar (up to noise) dynamics (and hence training time) as long as

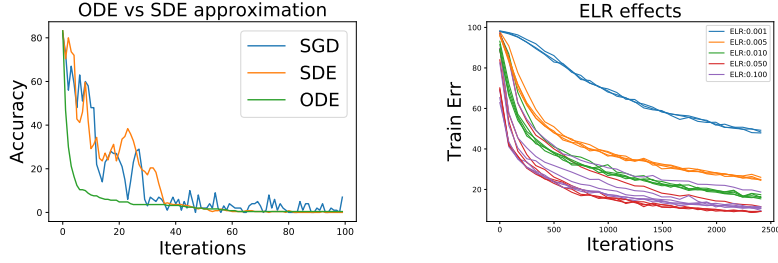

Figure 2: **(Left) ODE vs. SDE.** ODE approximation may not be well suited to describe the actual non-linear SGD dynamics (high learning rates regime). **(Right) Fine-tuning with the same ELR have similar curves**. We fine-tune an ImageNet pre-trained network on MIT-67 with different combinations of learning rates and momentum coefficients. We note that as long as the effective learning rate is the same, the loss curves are also similar.

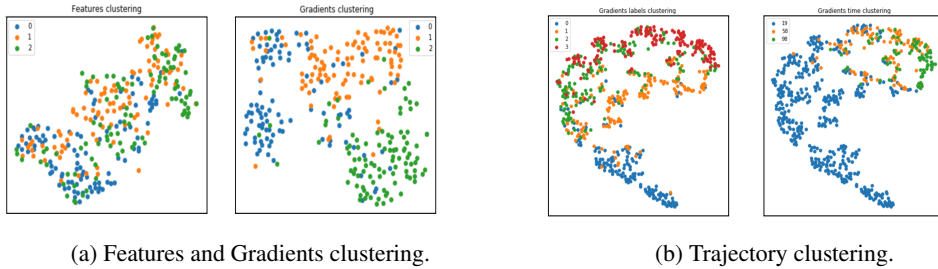

(a) Features and Gradients clustering.        (b) Trajectory clustering.

Figure 3: **Are gradients good descriptors to cluster data by semantics and training time?** (a) **Features vs Gradients clustering. (Right)** t-SNE plot of the first five principal components of the gradients of each sample in a subset of CIFAR-10 with 3 classes. Colors correspond to the sample class. We observe that the first 5 principal components are enough to separate the data by class. By Proposition 2 this implies faster training time. (**Left**) In the same setting as before, t-SNE plot of the features using the first 5 components of PCA. We observe that gradients separate the classes better than the features. **(b) t-SNE on predicted trajectories** To see if gradients are good descriptors of both semantics and training time we use gradients to predict linearized trajectories: we cluster the trajectories using t-SNE and we color each point by **(left)** class and **(right)** training time. We observe that: clusters split trajectories according both to labels **(left)** and training time **(right)**. Interestingly inside each class there are clusters of points that may converge at different speed.

the effective learning rate $\hat{\eta}$ remains the same. In Figure 2 we show empirically that indeed same effective learning rate implies similar loss curve. That similar effective learning rate gives similar test performance has also been observed in [22, 29].

**Batch size.** The batch size appears only in the stochastic part of the equation, its main effect is to decrease the scale of the SDE noise term. In particular, when the batch size goes to infinity $|B| \to \infty$ we recover the deterministic gradient flow also studied by [21]. Note that we need the batch size $|B|$ to go to infinity, rather than being as large as the dataset since we assumed random batch sampling with replacement. If we assume extraction without replacement the stochasticity is annihilated as soon as $|B| = N$ (see [7] for a more in depth discussion).

## 3.2 Effect of pre-training on training time

We now use the SDE in eq. (1) to analyze how the combination of different pre-trainings of the model – that is, different $w_0$'s – and different tasks affect the training time. In particular, we show that a necessary condition for fast convergence is that the gradients after pre-training cluster well with respect to the labels. We conduct this analysis for a binary classification task with $y_i = \pm 1$, but the extension is straightforward for multi-class classification, under the simplifying assumptions that we are operating in the limit of large batch size (GD) so that only the deterministic part of eq. (1) remains. Note the infinite batch size assumption is used only here to derive eq. (4), we make no such assumption in eq. (1), which is what we use for the actual training time prediction.

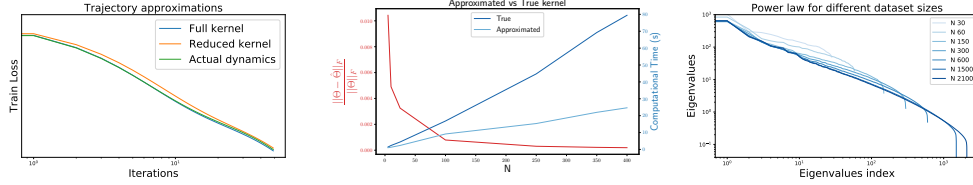

Figure 4: **(Left)** Actual fine-tuning of a DNN with GD compared to the numerical solution of eq. (1) and the solution using an approximated $\Theta$. The approximated $\Theta$ can faithfully describe fine-tuning dynamics while being twice as fast to compute and 100 times smaller to be stored. **(Center)** Relative difference in Frobenius norm of the real and approximated $\Theta$ as the dataset size varies (red), and their computational time (blue). **Right**: Eigen-spectrum of $\Theta$ computed on subsets of MIT-67 of increasing size. Note the convergence to a common power law (i.e., a line in log-log scale).

Under these assumptions, eq. (1) can be solved analitically and the loss of the linearized model at time $t$ can be written in closed form as (see Supplementary Material):

$$L_t = (\mathcal{Y} - f_0(\mathcal{X}))^T e^{-2\eta \Theta t} (\mathcal{Y} - f_0(\mathcal{X})) \tag{4}$$

The following characterization can easily be obtained using an eigen-decomposition of the matrix $\Theta$.

**Proposition 2** *Let $S = \nabla_w f_w(\mathcal{X})^T \nabla_w f_w(\mathcal{X})$ be the second moment matrix of the gradients and let $S = U \Sigma U^T$ be the uncentered PCA of the gradients, where $\Sigma = \mathrm{diag}(\lambda_1, \ldots, \lambda_n, 0, \ldots, 0)$ is a $D \times D$ diagonal matrix, $n \leq \min(N, D)$ is the rank of $S$ and $\lambda_i$ are the eigenvalues sorted in descending order. Then we have*

$$L_t = \sum_{k=1}^{D} e^{-2\eta \lambda_k t} (\delta \mathbf{y} \cdot \mathbf{v}_k)^2, \tag{5}$$

*where $\lambda_k \mathbf{v}_k = (g_i \cdot \mathbf{u}_k)_{i=1}^{N}$ is the $N$-dimensional vector containing the value of the $k$-th principal component of gradients $g_i$ and $\delta \mathbf{y} := \mathcal{Y} - f_0(\mathcal{X})$.*

**Training speed and gradient clustering.** We can give the following intuitive interpretation: consider the gradient vector $g_i$ as a representation of the sample $x_i$. If the first principal components of $g_i$ are sufficient to separate the classes (i.e., cluster them), then convergence is faster (see Figure 3). Conversely, if we need to use the higher components (associated to small $\lambda_k$) to separate the data, then convergence will be exponentially slower. Arora et al. [3] also use the eigen-decomposition of $\Theta$ to explain the slower convergence observed for a randomly initialized two-layer network trained with random labels. This is straightforward since the projection of a random vector will be uniform on all eigenvectors, rather than concentrated on the first few, leading to slower convergence. However, we note that the exponential dynamics predicted by [3] do not hold for more general networks trained from scratch [31] (see Section 6). In particular, eq. (5) mandates that the loss curve is always convex (it is sum of convex functions), which may not be the case for deep networks trained from scratch.

## 4 Efficient numerical estimation of training time

In Proposition 2 we have shown a closed form solution to the SDE in eq. (1) in the limit of large batch size, and for the MSE loss. Unfortunately, in general eq. (1) does not have a closed form expression when using the cross-entropy loss [21]. A numerical solution is however possible, enabled by the fact that we describe the network training in function space, which is much smaller than weight space for over-parametrized models. The main computational cost is to create the matrix $\Theta$ in eq. (1) – which has cost $O(DC^2 N^2)$ – and to compute the noise in the stochastic term. Here we show how to reduce the cost of $\Theta$ to $O(D_0 C^2 N^2)$ for $D_0 \ll D$ using a random projection approximation. Then, we propose a fast approximation for the stochastic part. Finally, we describe how to reduce the cost in $N$ by using only a subset $N' < N$ of samples to predict training time.

**Random projection.** To keep the notation uncluttered, here we assume w.l.o.g. $C = 1$. In this case the matrix $\Theta$ contains $N^2$ pairwise dot-products of the gradients (a $D$-dimensional vector) for each

of the $N$ training samples (see eq. 2). Since $D$ can be very large (in the order of millions) storing and multiplying all gradients can be expensive as $N$ grows. Hence, we look at a dimensionality reduction technique. The optimal dimensionality reduction that preserves the dot-product is obtained by projecting on the first principal components of SVD, which however are themselves expensive to obtain. A simpler technique is to project the gradients on a set of $D'$ standard Gaussian random vectors: it is known that such random projections preserve (in expectation) pairwise product [5, 1] between vectors, and hence allow us to reconstruct the Gram matrix while storing only $D'$-dimensional vector, with $D' \ll D$. We further increase computational efficiency using multinomial random vectors $\{-1,0,+1\}$ as proposed in [1] which further reduce the computational cost by avoiding floating point multiplications. In Figure 4 we show that the entries of $\Theta$ and its spectrum are well approximated using this method, while the computational time becomes much smaller.

**Computing the noise.** The noise covariance matrix $\Sigma$ is a $D \times D$-matrix that changes over time. Both computing it at each step and storing it is prohibitive. Estimating $\Sigma$ correctly is important to describe the dynamics of SGD [8], however we claim that a simple approximation may suffice to describe the simpler dynamic in function space. We approximate $\nabla_w f_0^{\mathrm{lin}}(\mathcal{X})\Sigma^{1/2}$ approximating $\Sigma$ with its diagonal (so that the we only need to store a $D$-dimensional vector). Rather than computing the whole $\Sigma$ at each step, we estimate the value of the diagonal at the beginning of the training. Then, by exploiting eq. (3), we see that the only change to $\Sigma$ is due to $\nabla_{f_t^{\mathrm{lin}}}\mathcal{L}$, whose norm decreases over time. Therefore we use the easy-to-compute $\nabla_{f_t^{\mathrm{lin}}}\mathcal{L}$ to re-scale our initial estimate of $\Sigma$.

**Larger datasets.** In the MSE case from eq. (4), knowing the eigenvalues $\lambda_k$ and the corresponding residual projections $p_k = (\delta\mathbf{y}\cdot\mathbf{v}_k)^2$ we can predict in closed form the whole training curve. Is it possible to predict $\lambda_k$ and $p_k$ using only a subset of the dataset? It is known [28] that the eigenvalues of the Gram matrix of Gaussian data follow a power-law distribution of the form $\lambda_k = ck^{-s}$. Moreover, by standard concentration argument, one can prove that the eigenvalues should converge to a given limit as the number of datapoints increases. We verify that a similar power-law and convergence result also holds for real data (see Figure 4). Exploiting this result, we can estimate $c$ and $s$ from the spectrum computed on a subset of the data, and then predict the remaining eigenvalues. A similar argument holds for the projections $p_k$, which also follow a power-law (albeit with slower convergence). We describe the complete estimation in the Supplementary Material.

## 5 Results

We now empirically validate the accuracy of proposition 1 in approximating the loss curve of an actual deep neural network fine-tuned on a large scale dataset. We also validate the goodness of the numerical approximations described in Section 4. Due to the lack of a standard and well established benchmark to test Training Time estimation algorithms we developed one with the main goal to closely resemble fine-tuning common practice for a wide spectrum of different tasks.

Table 1: Training Time absolute errors (number of steps) for CE loss using GD for $T = 150$ epochs at different thresholds $\epsilon$. TT estimates when ODE assumptions do and do not hold: high LR (0.005) and small LR (0.0001).

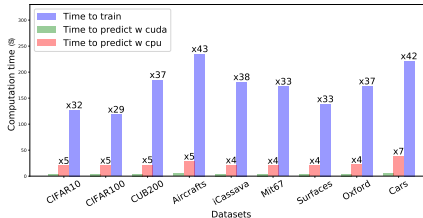

| TT error (# of steps) | $\epsilon = 1\%$ | | $\epsilon = 10\%$ | | $\epsilon = 40\%$ | |
|---|---|---|---|---|---|---|
| Lr | low | high | low | high | low | high |
| Cars [18] | 9 | 18 | 7 | 8 | 1 | 0 |
| Surfaces [4] | 6 | 13 | 6 | 7 | 6 | 3 |
| Mit67 [26] | 8 | 10 | 6 | 8 | 3 | 1 |
| Aircrafts [24] | 5 | 21 | 5 | 4 | 9 | 7 |
| CUB200 [30] | 6 | 6 | 5 | 8 | 1 | 1 |
| CIFAR100 [19] | 10 | 15 | 6 | 7 | 2 | 3 |
| CIFAR10 [19] | 9 | 14 | 8 | 9 | 3 | 3 |

Figure 5: Wall clock time (in seconds) to compute TT estimate vs fine-tuning running time. We run the methods described in Section 4 both on GPU and CPU. Training is done on GPU.

**Experimental setup.** We define training time as the first time the (smoothed) loss is below a given threshold. However, since different datasets converge at different speeds, the same threshold can be

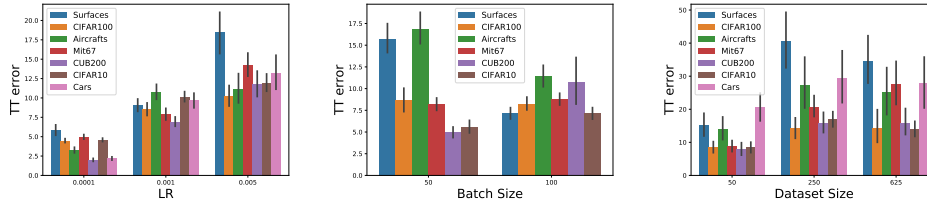

Figure 6: Average and 95% confidence intervals of TT estimate error for: **Left**: GD using different learning rates. **Center**: SGD using different batch sizes. **Right**: SGD using different dataset sizes. The average is taken w.r.t. random classes with different number of samples: {10, 50, 125}

too high (it is hit immediately) for some datasets, and too low for others (it may take hundreds of epochs to be reached). To solve this, and have cleaner readings, we define a 'normalized' threshold as follows: we fix the total number of fine-tuning steps $T$, and measure instead the first time the loss is within $\epsilon$ from the final value at time $T$. This measure takes into account the 'asymptotic' loss reached by the DNN within the computational budget (which may not be close to zero if the budget is low), and naturally adapts the threshold to the difficulty of the dataset. We compute both the real loss curve and the predicted training curve using Proposition 1 and compare the $\epsilon$-training-time measured on both. We report the *absolute prediction error*, that is $|t_{\text{predicted}} - t_{\text{real}}|$. For all the experiments we extract 5 random classes from each dataset (Table 1) and sample 150 images (or the maximum available for the specific dataset). Then we fine-tuned ResNet18/34 using either GD or SGD.

**Accuracy of the prediction.** In Figure 1 we show TT estimates errors (for different $\epsilon \in \{1, ..., 40\}$) under a plethora of different conditions ranging from different learning rates, batch sizes, datasets and optimization methods. For all the experiments we choose a multi-class classification problem with Cross Entropy (CE) Loss unless specified otherwise, and fixed computational budget of $T = 150$ steps both for GD and SGD. We note that our estimates are consistently within respectively a 13% and 20% relative error around the actual training time 95% of the times.

In Table 1 we describe the sensitivity of our estimates to different thresholds $\epsilon$ both when our assumptions do and do not hold (high and low learning rates regimes). Note that a larger threshold $\epsilon$ is hit during the initial convergence phase of the network, when a small number of iterations corresponds a large change in the loss. Correspondingly, the hitting time can be measured more accurately and our errors are lower. A smaller $\epsilon$ depends more on correct prediction of the slower asymptotic phase, for which exact hitting time is more difficult to estimate. See appendix C for a more in-depth discussion on both the error rate and the error distribution (underestimation vs overestimation) of our predictions.

**Wall-clock run-time.** In Figure 5 we show the wall-clock runtime of our training time prediction method compared to the time to actually train the network for $T$ steps. Our method is 30-40 times faster. Moreover, we note that it can be run completely on CPU without a drastic drop in performance. This allows to cheaply estimate TT and allocate/manage resources even without access to a GPU.

**Effect of dataset distance.** We note that the average error for Surfaces (Figure 6) is uniformily higher than the other datasets. This may be due to the texture classification task being quite different from ImageNet, on which the network is pretrained. In this case we can expect that the linearization assumption is partially violated since the features must adjust more during fine-tuning.

**Effect of hyper-parameters on prediction accuracy.** We derived Proposition 1 under several assumptions, importantly: small learning rate and $w_t$ close to $w_0$. In Figure 6 (left) we show that indeed increasing the learning rate decreases the accuracy of our prediction, albeit the accuracy remains good even at larger learning rates. Fine-tuning on larger dataset makes the weights move farther away from the initialization $w_0$. In Figure 6 (right) we show that this slightly increases the prediction error. Finally, we observe in Figure 6 (center) that using a smaller batch size, which makes the stochastic part of Proposition 1 larger also slightly increases the error. This can be ascribed to the approximation of the noise term (Section 4). On the other hand, in Figure 2 (right) we see that the effect of momentum on a fine-tuned network is very well captured by the effective learning rate (Section 3.1), as long as the learning rate is reasonably small, which is the case for fine-tuning. Hence the SDE approximation is robust to different values of the momentum. In general, we note that even when our assumptions are not fully met training time can still be approximated with only a slightly

higher error. This suggest that point-wise proximity of the training trajectory of linear and real models is not necessary as long as their behavior (decay-rate) is similar (see also Supplementary Material).

# 6    Discussion and conclusions

We have shown that we can predict with a 13-20% accuracy the time that it will take for a pre-trained network to reach a given loss, in only a small fraction of the time that it would require to actually train the model. We do this by studying the training dynamics of a linearized version of the model – using the SDE in eq. (1) – which, being in the smaller function space compared to parameters space, can be solved numerically. We have also studied the dependency of training time from pre-training and hyper-parameters (Section 3.1), and how to make the computation feasible for larger datasets and architectures (Section 4).

While we do not necessarily expect a linear approximation around a random initialization to hold during training of a real (non wide) network, we exploit the fact that when using a pre-trained network the weights are more likely to remain close to initialization [25], improving the quality of the approximation. However, in the Supplementary Material we show that even when using a pre-trained network, the trajectories of the weights of linearized model and of the real model can differ substantially. On the other hand, we also show that the linearized model correctly predicts the *outputs* (not the weights) of the real model throughout the training, which is enough to compute the loss. We hypothesise that this is the reason why eq. (1) can accurately predict the training time using a linear approximation.

The procedure described so far can be considered as an open loop procedure meaning that, since we are estimating training time before any fine-tuning step is performed, we are not gaining any feedback from the actual training. How to perform training time prediction during the actual training, and use training feedback (e.g., gradients updates) to improve the prediction in real time, is an interesting future direction of research.

# Broader Impact

This paper does not address a particular task, or a particular dataset, but rather addresses the technical issue of how to predict time and compute resources. As such, we expect it will benefit individual researchers with limited computational resources, by allowing them to optimize  for maximum impact. It also goes towards better understanding of the functioning of deep networks, so in that sense it could be thought of as contributing to improved interpretability of deep learning, in a broad sense. At this time, we do not foresee negative impacts beyond inaccurate predictions for some tasks and their consequences, which are mainly in the realm of waste of resources rather than changed outcomes. We have observed that our method yield predictions that have lower accuracy on some tasks rather than others, for instance it has lower accuracy on texture-based tasks than object classification. However, since we consider datasets as a whole, prediction inaccuracies do not impact any particular cohort or segment of the data.

# Acknowledgments and Disclosure of Funding

We would like to thank the anonymous reviewers for their feedback and suggestions.

The author(s) received no financial support for the research, authorship, and/or publication of this article.

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
