[Supplementary Material]

# Predicting Training Time Without Training: Supplementary Material

In the Supplementary Material we give the pseudo-code for the training time prediction algorithm (Appendix A) together with implementation details, show additional results including prediction of training time using only a subset of samples, and comparison of real and predicted loss curves in a variety of conditions (Appendix C). Finally, we give proofs of all statements.

## A    Algorithm

---
**Algorithm 1:** Estimate the Training Time on a given target dataset and hyper-parameters.

---
1: **Data:** Number of steps $T$ to simulate, threshold $\epsilon$ to determine convergence, pre-trained weights $w_0$ of the model, a target dataset with images $\mathcal{X} = \{x_i\}_{i=1}^N$ and labels $\mathcal{Y} = \{y_i\}_{i=1}^N$, batch size $B$, learning rate $\eta$, momentum $m \in [0,1)$.
2: **Result:** An estimate $\hat{T}_\epsilon$ of the number of steps necessary to converge within $\epsilon$ to the final value $T_\epsilon := \min\{t : |\mathcal{L}_t - \mathcal{L}_T| < \epsilon\}$.
3: **Initialization:** Compute initial network predictions $f_0(\mathcal{X})$, estimate $\Theta$ using random projections (Section 4), compute the ELR $\tilde{\eta} = \eta/(1-m)$ to use in eq. (1) instead of $\eta$;
4: **if** B = N **then**
5:     Get $f_t^{\mathrm{lin}}(\mathcal{X})$ solving the ODE in eq. (1) (only the *deterministic part*) for $T$ steps;
6: **else**
7:     Get $f_t^{\mathrm{lin}}(\mathcal{X})$ solving the SDE in eq. (1) for $T$ steps (see approximation in Section 4);
8: **end if**
9: Using $f_t^{\mathrm{lin}}(\mathcal{X})$ and $\mathcal{Y}$ compute linearized loss $\mathcal{L}_t^{lin} \quad \forall t \in \{1, ..., T\}$
10: **return** $\hat{T}_\epsilon := \min\{t : |\mathcal{L}_t^{lin} - \mathcal{L}_T^{lin}| < \epsilon\}$;

---

We can compute the estimate on training time based also on the accuracy of the model: we straight-forwardly modify the above algorithm and use the predictions $f_t^{\mathrm{lin}}(\mathcal{X})$ to compute the error instead of the loss (e.g. fig. 10).

We now briefly describe some implementations details regarding the numerical solution of ODE and SDE. Both of them can be solved by means of standard algorithms: in the ODE case we used LSODA (which is the default integrator in *scipy.integrate.odeint*), in the SDE case we used Euler-Maruyama algorithm for Ito equations.

We observe removing batch normalization (preventing the statistics to be updated) and removing data augmentation improve linearization approximation both in the case of GD and SGD. Interestingly data augmentation only marginally alters the spectrum of the Gram matrix $\Theta$ and has little impact on the linearization approximation w.r.t. batch normalization. [12] observed similar effects but, differently from us, their analysis has been carried out using randomly initialized ResNets.

## B    Target datasets

| Dataset | Number of images | Classes | Mean samples per class | Imbalance factor |
|---|---|---|---|---|
| cifar10 [19] | 50000 | 10 | 5000 | 1 |
| cifar100 [19] | 50000 | 100 | 500 | 1 |
| cub200 [30] | 5994 | 200 | 29.97 | 1.03 |
| fgvc-aircrafts [24] | 6667 | 100 | 66.67 | 1.02 |
| mit67 [26] | 5360 | 67 | 80 | 1.08 |
| opensurfacesminc2500 [4] | 48875 | 23 | 2125 | 1.03 |
| stanfordcars [18] | 8144 | 196 | 41.6 | 2.83 |

Table 2: Target datasets.

## C  Additional Experiments

**Prediction of training time using a subset of samples.** In Section 4 we suggest that in the case of MSE loss, it is possible to predict the training time on a large dataset using a smaller subset of samples (we discuss the details in Appendix D). In Figure 7 we show the result of predicting the loss curve on a dataset of $N = 4000$ samples using a subset of $N = 1000$ samples. Similarly, in Figure 11 (top row) we show the more difficult example of predicting the loss curve on $N = 1000$ samples using a very small subset of $N_0 = 100$ samples. In both cases we correctly predict that training on a larger dataset is slower, in particular we correctly predict the asymptotic convergence phase. Note in the case $N_0 = 100$ the prediction is less accurate, this is in part due to the eigenspectrum of $\Theta$ being still far from its limiting behaviour achieved for large number of data (see Appendix D).

**Comparison of predicted and real error curve.** In Figure 8 we compare the error curve predicted by our method and the actual train error of the model as a function of the number of optimization steps. The model is trained on a subset of 2 classes of CIFAR-10 with 150 samples. We run the comparison for both gradient descent (left) and SGD (right), using learning rate $\eta = 0.001$, momentum $m = 0$ and (in the case of SGD) batch size 100. In both cases we observe that the predicted curve is reasonably close to the actual curve, more so at the beginning of the training (which is expected, since the linear approximation is more likely to hold). We also perform an ablation study to see the effect of different approximation of SGD noise in the SDE in eq. (1). In Figure 8 (center) we estimate the variance of the noise of SGD at the beginning of the training, and then assume it is constant to solve the SDE. Notice that this predicts the wrong asymptotic behavior, in particular the predicted error does not converge to zero as SGD does. In Figure 8 (right) we rescale the noise as we suggest in Section 4: once the noise is rescaled the SDE is able to predict the right asymptotic behavior of SGD.

**Prediction accuracy in weight space and function space.** In Section 3 and Section 6 we argue that using a differential equation to predict the dynamics in function space rather than weight space is not only faster (in the over-parametrized case), but also more accurate. In Figure 9 we show empirically that solving the corresponding ODE in weight space leads to a substantially larger prediction error.

**Effective learning rate.** In Section 3.1 we note that as long as the effective learning rate $\tilde{\eta} = \eta/(1 - m)$ remains constant, runs with different learning rate $\eta$ and momentum $m$ will have similar learning curve. We show a formal derivation in Appendix E. In Figure 12 we show additional experiments, similar to Figure 2, on several other datasets to further confirm this point.

**Point-wise similarity of predicted and observed loss curve.** In some cases, we observe that the predicted and observed loss curves can differ. This is especially the case when using cross-entropy loss (Figure 10). We hypothesize that this may be due to improper prediction of the dynamics when the softmax output saturates, as the dynamic becomes less linear [21]. However, the train error curve (which only depends on the relative order of the outputs) remains relatively correct. We should also notice that prediction of the $\epsilon$-training-time $\hat{T}_\epsilon$ can be accurate even if the curves are not point-wise close. The $\epsilon$-training-time seeks to find the first time after which the loss or the error is within an $\epsilon$ threshold. Hence, as long as the real and predicted loss curves have a similar asymptotic slope the prediction will be correct, as we indeed verify in Figure 10 (bottom).

Figure 7: **Training-time prediction using a subset of the data. (Left)** Using the method described in Appendix D, we predict (green) the loss curve on a large dataset of $N = 4000$ samples (orange) using a subset of $N_0 = 1000$ samples (blue). In Figure 11 we show a similar result using a much smaller subset of $N_0 = 100$ samples. **(Right)** Corresponding estimated training time on the larger dataset at different thresholds $\epsilon$ compared to the real training time on the larger dataset.

Figure 8: **(Left)** Comparison of the real error curve on CIFAR10 using gradient descent and the predicted curve. **(Center)** Same as before, but this time we train using SGD and compare it with the prediction using the technique described in Section 4 to approximate the covariance of the SGD noise that appears in the SDE in eq. (1). **(Right)** Same as (center), but using constant noise instead of rescaling the noise using the value of the loss function as described in Section 4. Note that in this case we do not capture the right asymptotic behavior of SGD.

**Overestimation vs underestimation error**. Up to now we focused on prediction error rates (see e.g. table 1 and fig. 6), in this section we will briefly describe how the errors are distributed: are we overestimating or underestimating training time with our method? In Figure 10 we compare training time predictions and actual training time values for different thresholds $\epsilon$ and datasets. While for high thresholds $\epsilon$ the errors distribution does not seem to be particularly skewed (consider both MIT-67 and CIFAR-10) the situation is different for small thresholds $\epsilon$: our method tends to slightly overestimate training time. In particular overestimation is due to the non-linearity and high capacity of the network which might not be entirely captured by the linearization approximation (the non-linear model decreases its loss faster w.r.t. its linear approximation).

**Different training time definitions**: Our method to predict training time can be extended to other training time definitions. We started defining training time as the first time the (smoothed) loss is below a given threshold (which we then normalized w.r.t. the total computational budget allowed, see section 5). Similarly one can define training time as the first time training error decreases by less than $\epsilon$ over the last T epochs. With this definition we achieve 13% avg. relative prediction error and 0.94 Pearson's correlation between predicted and ground-truth time in the same experimental setting we used in section 5.

# D    Prediction of training time on larger datasets

In Section 4 we suggest that, in the case of MSE loss, it is possible to predict the training time on a large dataset using a subset of the samples. To do so we leverage the fact that the eigenvalues of $\Theta$ follows a power-law which is independent on the size of the dataset for large enough sizes (see Figure 4, right). More precisely, from Proposition 2, we know that given the eigenvalues $\lambda_k$ of $\Theta$ and

Figure 9: **Comparison of prediction accuracy in weight space vs. function space.** We compare the result of using the deterministic part of eq. (1) to predict the weights $w_t$ at time $t$ and the outputs $f_t(\mathcal{X})$ of the networks under GD. The relative error in predicting the outputs is much smaller than the relative error of predicting the weights at all times. This, together with the computational advantage, motivates the decision of using eq. (1) to predict the behavior in function space.

Figure 10: **Training time prediction is accurate even if loss curve prediction is not. (Top row)** Loss curve and error curve prediction on MIT-67 (left) and CIFAR-10 (right). **(Bottom row)** Predicted time to reach a given threshold (orange) vs real training time (blue). We note that on some datasets our loss curve prediction differs from the real curve near convergence. However, since our training time definition measures the time to reach the asymptotic value (which is what is most useful in practice) rather than the time reach an absolute threshold, this does not affect the accuracy of the prediction (see Appendix C).

the projections $p_k = \delta \mathbf{y} \cdot \mathbf{v}_k$ it is possible to predict the loss curve using

$$L_t = \sum_k p_k e^{-2\eta\lambda_k t}.$$

Let $\Theta_0$ be the Gram-matrix of the gradients computed on the small subset of $N_0$ samples, and let $\Theta$ be the Gram-matrix of the whole dataset of size $N$. Using the fact that, as we increase the number of samples, the eigenvalues (once normalized by the dataset size) converge to a fixed limit (Figure 4, right), we estimate the eigenvalues $\lambda_k$ of $\Theta$ as follow: we fit the coefficients $s$ and $c$ of a power law $\lambda_k = ck^{-s}$ to the eigenvalues of $\Theta_0$, and use the same coefficients to predict the eigenvalues of $\Theta$. However, we notice that the coefficient $s$ (slope of the power law) estimated using a small subset of the data is often smaller than the slope observed on larger datase (note in Figure 4 (right) that the curves for smaller datasets are more flat). We found that using the following corrected power law increases the precision of the prediction:

$$\hat{\lambda}_k = ck^{-s+\alpha\left(\frac{N_0}{N}-1\right)}.$$

Empirically, we determined $\alpha \in [0.1, 0.2]$ to give a good fit over different combinations of $N$ and $N_0$. In Figure 11 (center) we compare the predicted eigenspectrum of $\Theta$ with the actual eigenspectrum of $\Theta$ .

The projections $p_k$ follow a similar power-law – albeit more noisy (see Figure 11, right) – so directly fitting the data may give an incorrect result. However, notice that in this case we can exploit an additional constraint, namely that $\sum_k p_k = \|\delta \mathbf{y}\|^2$ ($\|\delta \mathbf{y}\|^2$ is a known quantity: labels and initial model predictions on the large dataset). Let $p_k = \delta \mathbf{y} \cdot \mathbf{v}_k$ and let $p'_k = \delta \mathbf{y} \cdot \mathbf{v}'_k$ where $\mathbf{v}_k$ and $\mathbf{v}'_k$ are the eigenvectors of $\Theta$ and $\Theta_0$ respectively. Fix a small $k_0$ (in our experiments, $k_0 = 100$). By convergence laws [28], we have that $p'_k \simeq p_k$ when $k < k_0$. The remaining tail of $p_k$ for $k > k_0$ must now follow a power-law and also be such that $\sum_k p_k = \|\delta \mathbf{y}\|^2$. This uniquely identify the coefficients of a power law. Hence, we use the following prediction rule for $p_k$:

$$\hat{p}_k = \begin{cases} p'_k & \text{if } k < k_0 \\ ak^{-b} & \text{if } k \geq k_0 \end{cases}$$

where $a$ and $b$ are such that $\hat{p}_{k_0} = p'_{k_0}$ and $\sum_k \hat{p}_k = \|\delta \mathbf{y}\|^2$.

In Figure 11 (left), we use the approximated $\hat{\lambda}_k$ and $\hat{p}_k$ to predict the loss curve on a dataset of $N = 1000$ samples using a smaller subset of $N_0 = 100$ samples. Notice that we correctly predict that the convergence is slower on the larger dataset. Moreover, while training on the smaller dataset quickly reaches zero, we correctly estimate the much slower asymptotic phase on the larger dataset. Increasing both $N$ and $N_0$ increases the accuracy of the estimate, since the eigenspectrum of $\Theta$ is closer to convergence: In Figure 7 we show the same experiment as Figure 11 with $N_0 = 1000$ and $N = 4000$. Note the increase in accuracy on the predicted curve.

Figure 11: **Training time prediction using a subset of the data. (Left)** We predict the loss curve on a large dataset of $N = 1000$ samples using a subset of $N_0 = 100$ samples on CIFAR10 (similar results hold for other datasets presented so far). **(Center)** Eigenspectrum of $\Theta$ computed using $N_0 = 100$ samples (orange), $N = 1000$ samples (green) and predicted spectrum using our method (blue). **(Right)** Value of the projections $p_k$ of $\delta\mathbf{y}$ on the eigenvectors of $\Theta$, computed at $N_0 = 100$ (orange) and $N = 1000$ (blue). Note that while they approximatively follow a power-law on average, it is much more 'noisy' than that of the eigenvalues. In green we show the predicted trend using our method.

## E   Effective learning rate

We now show that having a momentum term has the effect of increasing the effective learning rate in the deterministic part of eq. (1). A similar treatment of the momentum term is also in [29, Appendix D]. Consider the update rule of SGD with momentum:

$$a_{t+1} = m\,a_t + g_{t+1},$$
$$w_{t+1} = w_t - \eta\,a_{t+1},$$

If $\eta$ is small, the weights $w_t$ will change slowly and we can consider $g_t$ to be approximately constant on short time periods, that is $g_{t+1} = g$. Under these assumptions, the gradient accumulator $a_t$ satisfies the following recursive equation:

$$a_{t+1} = m\,a_t + g,$$

which is solved by (assuming $a_0 = 0$ as common in most implementations):

$$a_t = (1 - m^t)\frac{g}{1 - m}.$$

In particular, $a_t$ converges exponentially fast to the asymptotic value $a^* = g/(1 - m)$. Replacing this asymptotic value in the weight update equation above gives:

$$w_{t+1} = w_t - \eta a^* = w_t - \frac{\eta}{1 - m}g = w_t - \tilde{\eta}\,g,$$

that is, once $a_t$ reaches its asymptotic value, the weights are updated with an higher effective learning rate $\tilde{\eta} = \frac{\eta}{1-m}$. Note that this approximation remains true as long as the gradient $g_t$ does not change much in the time that it takes $a_t$ to reach its asymptotic value. This happens whenever the momentum $m$ is small (since $a_t$ will converge faster), or when $\eta$ is small ($g_t$ will change more slowly). For larger momentum and learning rate, the effective learning rate may not properly capture the effect of momentum.

## F   Proof of theorems

### F.1   Proposition 1: SDE in function space for linearized networks trained with SGD

We now prove our Proposition 1 and show how we can approximate the SGD evolution in function space rather than in parameters space. We follow the standard method used in [13] to derive a general SDE for a DNN, then we speciaize it to the case of linearized deep networks. Our notation follows [21], we define $f_{\theta_t}(\mathcal{X}) = vec([f_t(x)]_{x\in\mathcal{X}}) \in \mathbb{R}^{CN}$ the stacked vector of model output logits for all examples, where $C$ is the number of classes and $N$ the number of samples in the training set.

To describe SGD dynamics in function space we start from deriving the SDE in parameter space. In order to derive the SDE required to model SGD we will start describing the discrete update of SGD as done in [13].

$$\theta_{t+1} = \theta_t - \eta\nabla_\theta\mathcal{L}^B(\theta_t) \tag{6}$$

Figure 12: **Additional experiments on the effective learning rate.** We show additional plots showing the error curves obtained on different datasets using different values of the effective learning rate $\tilde{\eta} = \eta/(1 - m)$, where $\eta$ is the learning rate and $m$ is the momentum. Each line is the observed error curve of a model trained with a different learning rate $\eta$ and momentum $m$. Lines with the same color have the same ELR $\tilde{\eta}$, but each has a different $\eta$ and $m$. As we note in Section 3.1, as long as $\tilde{\eta}$ remains the same, training dynamics with different hyper-parameters will have similar error curves.

where $\mathcal{L}^B(\theta_t) = \mathcal{L}(f_{\theta_t}(\mathcal{X}^B), \mathcal{Y}^B)$ is the average loss on a mini-batch $B$ (for simplicity, we assume that $B$ is a set of indexes sampled with replacement).

The mini-batch gradient $\nabla_\theta \mathcal{L}^B(\theta_t)$ is an unbiased estimator of the full gradient, in particular the following holds:

$$\mathbb{E}[\nabla_\theta \mathcal{L}^B(\theta_t)] = 0 \qquad \text{cov}[\nabla_\theta \mathcal{L}^B(\theta_t)] = \frac{\Sigma(\theta_t)}{|B|} \qquad (7)$$

Where we defined the covariance of the gradients as:

$$\Sigma(\theta_t) := \mathbb{E}\big[(g_i \nabla_{f_t(x_i)} \mathcal{L}) \otimes (g_i \nabla_{f_t(x_i)} \mathcal{L})\big] - \mathbb{E}\big[g_i \nabla_{f_t(x_i)} \mathcal{L}\big] \otimes \mathbb{E}\big[g_i \nabla_{f_t(x_i)} \mathcal{L}\big]$$

and $g_i := \nabla_w f_0(x_i)$. The first term in the covariance is the second order moment matrix while the second term is the outer product of the average gradient.

Following standard approximation arguments (see [7] and references there in) in the limit of small learning rate $\eta$ we can approximate the discrete stochastic equation eq. (6) with the SDE:

$$d\theta_t = -\eta \nabla_\theta \mathcal{L}(\theta_t) dt + \frac{\eta}{\sqrt{|B|}} \Sigma(\theta_t)^{\frac{1}{2}} dn \qquad (8)$$

where $n(t)$ is a Brownian motion.

Given this result, we are going now to describe how to derive the SDE for the output $f_t(\mathcal{X})$ of the network on the train set $\mathcal{X}$. Using Ito's lemma (see [13] and references there in), given a random variable $\theta$ that evolves according to an SDE, we can obtain a corresponding SDE that describes the evolution of a function of $\theta$. Applying the lemma to $f_\theta(\mathcal{X})$ we obtain:

$$df_t(\mathcal{X}) = [-\eta \Theta_t \nabla_{f_t} \mathcal{L}(f_t(\mathcal{X}), \mathcal{Y}) + \frac{1}{2} vec(A)] dt + \frac{\eta}{\sqrt{|B|}} \nabla_\theta f(\mathcal{X}) \Sigma(\theta_t)^{\frac{1}{2}} dn \qquad (9)$$

where $\nabla_\theta f(\mathcal{X}) \in \mathbb{R}^{CN \times D}$ is the jacobian matrix and $D$ is the number of parameters. Note $A$ is a $N \times C$ matrix which, denoting by $f_\theta^{(j)}(x)$ the $j$-th output of the model on a sample $x$, is given by:

$$A_{ij} = \text{tr}[\Sigma(\theta_t) \nabla_\theta^2 f_\theta^{(j)}(x_i)].$$

Using the fact that in our case the model is linearized, so $f_\theta(x)$ is a linear function of $\theta$, we have that $\nabla_\theta^2 f^{(j)}(x) = 0$ and hence $A = 0$. This leaves us with the SDE:

$$df_t(\mathcal{X}) = -\eta \Theta_t \nabla_{f_t} \mathcal{L} dt + \frac{\eta}{\sqrt{|B|}} \nabla_\theta f(\mathcal{X}) \Sigma(\theta_t)^{\frac{1}{2}} dn \qquad (10)$$

as we wanted.

## F.2 Proposition 2: Loss decomposition

Let $\nabla_w f_w(\mathcal{X}) = V\Lambda U$ be the singular value decomposition of $\nabla_w f_w(\mathcal{X})$ where $\Lambda$ is a rectangular matrix (of the same size of $\nabla_w f_w(\mathcal{X})$) containing the singular values $\{\sigma_1, \ldots, \sigma_N\}$ on the diagonal. Both $U$ and $V$ are orthogonal matrices. Note that we have

$$S = \nabla_w f_w(\mathcal{X})^T \nabla_w f_w(\mathcal{X}) = U^T \Lambda^T \Lambda U,$$

$$\Theta = \nabla_w f_w(\mathcal{X}) \nabla_w f_w(\mathcal{X})^T = V\Lambda\Lambda^T V^T.$$

We now use the singular value decomposition to derive an expression for $\mathcal{L}_t$ in case of gradient descent and MSE loss (which we call $L_t$). In this case, the differential equation eq. (1) reduces to:

$$\dot{f}_t^{\text{lin}}(\mathcal{X}) = -\eta\Theta(\mathcal{Y} - f_t^{\text{lin}}(\mathcal{X})),$$

which is a linear ordinary differential equation that can be solved in closed form. In particular, we have:

$$f_t^{\text{lin}}(\mathcal{X}) = (I - e^{-\eta\Theta t})\mathcal{Y} + e^{-\eta\Theta t} f_0(\mathcal{X}).$$

Replacing this in the expression for the MSE loss at time $t$ we have:

$$\begin{aligned} L_t &= \sum_i (y_i - f_t^{\text{lin}}(x_i))^2 \\ &= (\mathcal{Y} - f_t^{\text{lin}}(\mathcal{X}))^T (\mathcal{Y} - f_t^{\text{lin}}(\mathcal{X})) \\ &= (\mathcal{Y} - f_0(\mathcal{X}))^T e^{-2\eta\Theta t} (\mathcal{Y} - f_0(\mathcal{X})). \end{aligned}$$

Now recall that, by the properties of the matrix exponential, we have:

$$e^{-2\eta\Theta t} = e^{-2\eta V\Lambda\Lambda^T V^T t} = V e^{-2\eta\Lambda\Lambda^T t} V^T,$$

where $e^{-2\Lambda\Lambda^T t} = \text{diag}(e^{-2\eta\lambda_1 t}, e^{-2\eta\lambda_2 t}, \ldots)$ with $\lambda_k := \sigma_k^2$. Then, defining $\delta\mathbf{y} = \mathcal{Y} - f_0(\mathcal{X})$ and denoting with $\mathbf{v}_k$ the $k$-th column of $V$ we have:

$$\begin{aligned} L_t &= \delta\mathbf{y}^T V e^{-2\eta\Lambda\Lambda^T t} V^T \delta\mathbf{y} \\ &= \sum_{k=1}^N e^{-2\eta\lambda_k t}(\delta\mathbf{y} \cdot \mathbf{v}_k). \end{aligned}$$

Now let $\mathbf{u}_k$ denote the $k$-th column of $U^T$ and $g_i$ the $i$-th column of $\nabla_w f_w(\mathcal{X})^T$ (that is, the gradient of the $i$-th sample). To conclude the proof we only need to show that $\lambda_k \mathbf{v}_k = (g_i \cdot \mathbf{u}_k)_{i=1}^N$. But this follows directly from the SVD decompostion $\nabla_w f_w(\mathcal{X}) = V\Lambda U$, since then $V\Lambda = \nabla_w f_w(\mathcal{X})U^T$.