[Reviews · NeurIPS 2020]

Review 1

Summary and Contributions: Authors have shown that it is possible to predict the number of optimization steps required to fine tune a pretrained deep network to converge to a given value of the loss function, with a 13-20% error margin and 95% confidence, at a 30 to 45-fold reduction in cost compared to actual training. To achieve this, authors assumed that the fine tuning process can be approximated by linearized DNN and modeled the SGD process using a Stochastic Differential Equation in function space rather than in weight space, which significantly reduces the computational cost. They also used random projection and sampling to reduce the cost further.

Strengths: The work is well referenced and theoretical grounding is sound. Experiments are well designed to show the effectiveness of the proposed method.

Weaknesses: My main concern is the significance of the contribution. As a practitioner, I think it is not too difficult to predict the # of SGD iterations a network takes to be fine-tuned, especially if i tried the same task a few times before. Also, it would not be useful if advanced optimization techniques were used.

Correctness: Yes, I think they are correct.

Clarity: Yes, it is very well written.

Relation to Prior Work: Yes.

Reproducibility: Yes

Additional Feedback:


Review 2

Summary and Contributions: This paper develops an approach to estimating the number of training steps of deep neural networks for a given fine-tuning task. The key idea is to use a linearized model to approximate the neural network and model its training dynamics by a stochastic differential equation (SDE). Under a set of assumptions, the solution of this SDE can be obtained to predict the evolution of loss value with respect to the number of training steps. Experimental study demonstrates the effectiveness of the proposed approach.

Strengths: 1. This paper addresses an issue that is of significance but has not been sufficiently researched at the current stage. 2. The proposed technique and approach are sound. 3. By giving a good estimate of the number of training steps, this research can help to better schedule network training and allocate computational resources. 4. This work is relevant to the NeurIPS community.

Weaknesses: 1. It seems that the key ideas and results of this work are mainly built upon the existing work in the literature such as [15], [14], [25], [13], etc. Therefore, the technical novelty and contributions of this work shall be made clearer with respect to these pieces of work. 2. Various assumptions and relaxations (say, the unrealistic infinity batch size) have to be taken to derive the result in Eq.(4). This seems to be a weakness for the effectiveness of this approach for practical tasks. At the same time, the experimental result demonstrates that the proposed method can work well for practical tasks, which is good.

Correctness: Yes. the claims and methods are overall sound and well explained.

Clarity: This paper is overall well written although it needs some efforts to understand the technical details.

Relation to Prior Work: As indicated above, this work can be strengthened at this aspect. Please better highlight the technical novelty and contributions of this work with respect to the existing work upon which the key ideas and results are built.

Reproducibility: Yes

Additional Feedback: 1. How does network architecture affect the effectiveness of the proposed method? Also, how do the type and number of network layers to be fine-tuned affect this method? Please comment. 2. In this work, random projection is used to reduce the dimensions of the high-dimensional gradient vectors from D dimensions to D'. Are these gradient vectors sparse enough? It is understood that usually random projection works effectively with sparse high-dimensional vectors. 3. In Table 1, the sensitivity of the estimates is presented. However, it is not clear what exactly the values in Table 1 represent. That is, how is the sensitivity represented by the value? Please clarify. 4. In the experiment, the total number of fine-tuning steps T and the threshold \epsilon are used to help decide the number of training steps. In practice, what information does a user need to provide for the proposed method to estimate the number of training steps? /------------------------------------------------------------- After reading the comments of peer reviewers and the author feedback, the score is maintained to be 7. This paper addresses an issue that is of practical significance with sound theoretical modelling and analysis. The experimental study also demonstrates the potential of the proposed method. The author feedback is clear and informative. This work can be further enhanced by incorporating some of the discussions provided in the author feedback.


Review 3

Summary and Contributions: This article presents a method for predicting the training time of a given neural network (NN) with given loss and dataset. The method is based on a theoretical analysis of pre-trained neural networks combined with results coming from the Neural Tangent Kernel (NTK) framework. On the experimental side, the authors provide several approximations in order to make their method computationally feasible. According to the experimental results, the proposed method predicts training time with an error comprised between 2.5% and 20%, depending on the dataset. EDIT: I have read authors' feedback, and I still have mixed feelings about this paper. The theoretical contribution is interesting, but is basically a combination of two previous works, and the pre-trained condition is very limiting. However, I believe that such theoretical approach of training time prediction would be interesting for the ML community.

Strengths: The authors explore a problem that has not been addressed (that is predicting the training time of a NN). Unlike works about “learning curve prediction” (which is close to the current problem), the authors exploit only theoretical results about NNs. This point is important, since we also need to test the validity of usual approximations in NN theory. By the way, the authors have tested some limits of these approximations (small learning rate).

Weaknesses: Predicting the training time of a NN which has already been pre-trained has very few applications.

Correctness: The proof of the main result is a simple combination of “Gradients as Features for Deep Representation Learning”, Mu 2020, and “Mean-field Behaviour of Neural Tangent Kernel for Deep Neural Networks”, Hayou 2019. Thus, it is easy to check. The authors propose a fair definition of “convergence time”, based on the number of epochs needed to bring the loss under a threshold epsilon*L_0, where epsilon is chosen by the user (and discussed by the authors) and L_0 is the loss after 150 epochs. The authors assume that, after 150 epochs, the NN has converged. It may be the case with their datasets and models, but this criterion would not hold for more complex tasks. I suggest, for instance, that training ends when the loss has not decreased by a factor epsilon over the last T epochs (where epsilon and T will be discussed and fixed). The authors only provide the error rate of their predictions, but do not indicate whether their method tends to overestimate or underestimate the training time. We can see some clues in the supplementary material, but this should appear in the paper.

Clarity: There is no major writing issue. Just one remark: the paper “Mean-field behaviour of neural tangent kernel for deep neural networks” has two entries in the bibliography instead of one.

Relation to Prior Work: To build their method, the authors assumed that the NN to be trained is in fact already pre-trained. This assumption justifies the “linear approximation”, which is absolutely necessary. As explained by the authors, this framework comes from the article “Gradients as Features for Deep Representation Learning”, Mu 2020. They have combined it with one theoretical result of the article “Mean-field Behaviour of Neural Tangent Kernel for Deep Neural Networks”, Hayou 2019, to obtain their theoretical result. Concerning the “prior works”, I expected that some methods for learning curve prediction would be cited (e.g. "Learning Curve Prediction with Bayesian Neural Networks", Klein 2017, contains a few references). To my knowledge, they are the closest works to the presented method, although they are not concurrent. At least, a short discussion between the actual goal of the authors and learning curve prediction should be presented. Moreover, the authors tried to perform learning curve prediction (but in the appendix only).

Reproducibility: Yes

Additional Feedback:


Review 4

Summary and Contributions: This paper proposes a way to predict the number of optimization steps that a pre- trained deep network needs to converge to a given value of the loss function. It assumes that the deep network during fine-tuning are approximated by those of a linearized model.

Strengths: It tackles an interesting problem, predicting the training time with different batch sizes and learning rates. It is tested on various datasets.

Weaknesses: The method assumes when a pre-trained network is fine-tuned, the solution remains close to the pre-trained weights which is not true for the datasets that are visually different and have a domain gap with the pre-training dataset. That's a big limitation of the work. With that limitation, the use case is limited. What are the accuracy of the trained models? Are they trained long enough to converge to their top accuracy? What are the accuracies of a network when trained long enough as in the real training time and when trained in predicted training time - is there a big gap? What is the dataset size in Figure 6 - is it 50, 250 and 625 images for training?

Correctness: The paper claims predicting training time. I am not sure if the predicted times match similar accuracies.

Clarity: The introduction starts by talking a linearized version of the DNN, I don't know what linearized version is. In Figure 1, is the predicted-real training times are plotted by increasing the dataset size, changing the batch size or learning rate? caption and figure don't say. What are the numbers in Table 1? Datasets are not explained, for example I don't know what surfaces dataset is, and how big it is.

Relation to Prior Work: The paper states that predicting the training time of a state-of-the-art architecture on large scale datasets is a relatively understudied topic. This paper is also the first one I read in this topic.

Reproducibility: Yes

Additional Feedback: I thank the authors for the rebuttal. After reading it and other reviewers comments, I increase my score to 6.

[Author Response · NeurIPS 2020]

We thank the reviewers for their feedback. We reply to the main points below:

**Usefulness of the method (R1, R3) and applicability of assumptions (R4):** Actually, this work started precisely to
unblock the adoption of a real-world Computer Vision AutoML system: Users fine-tune models selected from a large
model zoo testing hundreds of combinations of different architectures, pre-training sets and hyper-parameters, but are
reluctant to do so without visibility of the expected ROM cost of the training. We focus on fine-tuning (R3) since it is
faster and typically performs better than training from scratch in vision problems. For these reasons, it is generally the
choice in AutoML systems, where users pay by the hour. While our work may not directly impact academic researchers,
it is an enabler of large-scale AutoML, which we expect will further foster academic research in the years ahead. In this
sense, we would say our work is useful: it enables a cost estimate that allows reducing a large search space to fit a user's
budget, a small step toward better accessibility, and democratization of ML. We are happy to expand the discussion in
this direction if it is of interest to the readers.

Note the technical hypotheses that the weights remain close to initialization (R4) is only used to derive the approximation,
which is then verified empirically to hold (albeit with a larger error margin) even when the task is quite different from
pre-training (e.g., fine-grained airplane classification using ImageNet pre-training, see also Fig. 10 and Section B). In
real-world AutoML this concern is largely moot since model selection techniques selects for further fine-tuning only
models pre-trained on close data.

**Make contributions over related literature more clear (R2):** Our main contribution is to introduce the problem of
predicting training time in realistic use cases (see previous point), in particular how this depends on the hyper-parameters
(for which some previous literature exists) and on the interaction between target task and pre-training (which, to the best
of our knowledge, is new). *NTK theory* [15] studies randomly initialized DNNs in the limit of infinite width and batch
size. We replace it with an off-the-shelf pre-trained network fine-tuned with SGD and show that its predictions can be
effective on practical networks. We also show how to approximate the NTK matrix efficiently while maintaining a good
accuracy for our end-tasks. *Gradients as feature* [25] uses a linearization-based analysis of fine-tuning which is similar
to ours, but it focuses on efficiently using the gradients of the network as features for a linear classifier, and is unrelated
to our work in both scope and methods. *Mean field approximation* [13] focuses on describing a special initialization
technique. However, we build on their SDE approximation in weight space and show that it can be translated to function
space, making a numerical solution possible for real sized DNNs.

**Connection to learning curve prediction methods (R3):** We thank the reviewer for pointing out this relevant area of
research, which we will discuss in the paper. We should note that these papers focus on predicting the effect of different
hyper-parameters for fixed task and architecture. However, we found the relation between target task and pre-training is
more difficult to model. We initially employed a similar black-box regressions techniques to predict training time, but
they turned out to be data-hungry and less likely to generalize on different models. This prompted us to develop a more
interpretable and less data expensive approach presented here.

**Experiment with different definition of training time (R3), different architectures (R2)**: We further tested our
method with the suggested definition (first time training error decreases by less than $\epsilon$ in one epoch), and also increased
the number of iterations to 600 to ensure convergence and top accuracy. In this (more challenging) case, we achieve
13% avg. relative prediction error and 0.94 Pearson's correlation between predicted time and ground-truth. The method
also generalizes to several deeper architectures (ResNet-50, Densenet-121). However, we note that predictions for older
architectures (AlexNet) is less accurate (40% avg. error, 0.46 Pearson's correlation). This is expected as their loss
landscape is known to be very rough, so gradients at initialization are less informative.

**Clarifications**: We have amended the text to make all requested clarifications. More in detail:

• **Table 1 (R2, R4):** We report the absolute error on training-time prediction as a function of the selected threshold.
We show the error both using a small learning rate (our approximation holds better) and high learning rate. Different
thresholds reflect training time predictions at different regimes: initial (high $\epsilon$) and final (low $\epsilon$) convergence.

• **Random projections (R2):** We do not need the gradients to be sparse. We use the property of random projections to
preserve the expected L2 distances (and hence inner product) when applied to high dimensional vectors (see [1,5]).

• **Figure 1 (R4):** For each point we sample a different batch size, learning rate and a different dataset size.

• **Dataset details (R4):** We added a table describing for each dataset the number of images and classes.

• **Overestimation or underestimation? (R3):** We slightly overestimate the training time. This is due to the non-
linearity and high capacity of the network which is not entirely captured by the linearization approximation (see
Section B).

• **Infinite batch size assumption (R2):** Please note that this is used only in eq. (4) to give an analytical interpretation.
We make no such assumption in eq. (1), which is what we use for the actual training time prediction.

[Meta-Review · NeurIPS 2020]

All the reviewers unanimously agree that the paper should be accepted.